# Pyrolysis Study of Mixed Polymers for Non-Isothermal TGA: Artificial Neural Networks Application

**DOI:** 10.3390/polym14132638

**Published:** 2022-06-28

**Authors:** Ibrahim Dubdub

**Affiliations:** Department of Chemical Engineering, King Faisal University, Al-Hassa 31982, Saudi Arabia; idubdub@kfu.edu.sa

**Keywords:** pyrolysis, mixed polymers, thermogravimetric analyzer (TGA), artificial neural networks (ANN)

## Abstract

Pure polymers of polystyrene (PS), low-density polyethylene (LDPE) and polypropylene (PP), are the main representative of plastic wastes. Thermal cracking of mixed polymers, consisting of PS, LDPE, and PP, was implemented by thermal analysis technique “thermogravimetric analyzer (TGA)” with heating rate range (5–40 K/min), with two groups of sets: (ratio 1:1) mixture of PS and PP, and (ratio 1:1:1) mixture of PS, LDPE, and PP. TGA data were utilized to implement one of the machine learning methods, “artificial neural network (ANN)”. A feed-forward ANN with Levenberg-Marquardt (LM) as learning algorithm in the backpropagation model was performed in both sets in order to predict the weight fraction of the mixed polymers. Temperature and the heating rate are the two input variables applied in the current ANN model. For both sets, 10-10 neurons in logsig-tansig transfer functions two hidden layers was concluded as the best architecture, with almost (R > 0.99999). Results approved a good coincidence between the actual with the predicted values. The model foresees very efficiently when it is simulated with new data.

## 1. Introduction

Recently, most of the researchers are aiming to deal with machine learning methods “ANN” for the forecasting of different data since it approved that it has a strong performance to deal with non-linearity relationships. Therefore, ANN is considered as another option to deal with the TGA datum.

The literatures surveyed listed below will be limited only for the papers handling ANN for TGA data [1,2,3,4,5,6,7,8,9,10,11,12,13,14,15,16,17,18].

Conesa et al. [1] was the first to explore ANN in the thermal analysis by initiating a way to treat with the pyrolysis kinetics at different samples for non-isothermal runs. Bezerra et al. [2] applied the ANN model to the thermal cracking of carbon fiber/phenolic resin composite laminate. Yıldız et al. [3] examined the oxidation of mixtures of different ratio by enforcing ANN. Çepelioĝullar et al. [4] extended an ANN to foresee the pyrolysis of waste fuel. Ahmad et al. [5] established ANN for the pyrolysis of Typha latifolia. They collected 1021 data for the feed-forward Levenberg–Marquardt back-propagation algorithm. Çepelioĝullar et al. [6] performed the ANN models for Lignocellulosic forest residue (LFR) and olive oil residue (OOR) in two different sets: (i) two separate networks for each sample, and (ii) one network for both samples. Later, Chen et al. [7] studied the co-combustion characteristics of sewage sludge and coffee grounds (CG) mixtures. Naqvi et al. [8] suggested an ANN to tip the thermal cracking of one type of sludge and offered a strong harmonization for the predicted with experimental figures. In this paper, a richly powerful promoted ANN model (R ≈ 1.0) predicted a pyrolytic behavior of mixed polymers. Ahmad et al. [9] validated the pyrolysis of Staghorn Sumac by ANN model. 

Bi et al. [10] investigated the co-combustion co-pyrolysis of sewage sludge and peanut shell by ANN model. Bong et al. [11] applied the ANN model for the catalytic pyrolysis of pure microalgae, peanut shell wastes, and their binary mixtures with the microalgae ash as a catalyst. In addition, Bi et al. [12] repeated the study for the co-pyrolysis of coal gangue and peanut shell. In both papers, they found there was consistency between the experimental and the ANN model results. Liew et al. [13] predicted the co-pyrolysis of corn cob and high-density polyethylene (HDPE) mixtures, with chicken and duck egg shells as catalysts. Zaker et al. [14] investigated the effects of two catalysis (HZSM5 and sludge-derived activated char) on the pyrolysis of sewage sludge. Dubdub and Al-Yaari [15,16] and Al-Yaari and Dubdub [17,18] tried to use the ANN to predict the performance for different samples. They used a feed-forward LM optimization technique for backpropagation process in the ANN model, in two hidden layers. In the first paper, they applied two input variables, temperature and heating rate, and one output variable, weight left %, while in the second paper, catalyst/polymer weight ratio was added as third input. 

Almost all of the above-mentioned studies have good agreement between the experimental collected data and the ANN predicted results efficiently in common. The architecture details of all the papers above are similar to this work (non-isothermal TGA data) are summarized in Table 1. Most of these papers used the temperature and the heating rate for the input variables with weight left % as the only output. This table showed and approved that the application of ANN to predict TGA data is feasible and promising research. In this work, the novelty of this work is in applying the ANN for new two mixture of polymers (PS, LDPE, and PP), and using the final best architecture efficiently in the simulation of new input data. 

## 2. Materials and Methods

### 2.1. Thermal Decomposition 

Pyrolysis experiments were conducted under nitrogen with different compositions of three polymers: PP, PS, and LDPE. Table 2 shows six tests of two sets: tests 1–3 (ratio 1:1) binary of PS and PP, and tests 4–6 (ratio 1:1:1) of PS, LDPE, and PP. 10 mg of each powder sample was used throughout the study. Proximate and ultimate analysis that was performed to characterize the polymer samples can be found in reference [16]. Thermal decomposition experiments were conducted under N2 (99.999%) gas flowing at 100 cm^3^/min using the thermogravimetric analyzer (TGA-7), manufactured by PerkinElmer, Shelton, CT, USA [16].

### 2.2. Structure of ANNs

The common procedure for modelling engineering units is to develop a model depending on the basic principles of physics and chemistry and then the values of the model parameters are estimated from some experimental data by some numerical techniques. However, formulating any model and finding the values of the parameters are the most difficult works in most of the cases, especially when the final model is very complicated with non-linear relations among the variables. In these cases, the ANN may become the alternative option. One of the strengths of ANN is its ability to model the non-linear functions and complex process by mapping these relations by some approximation functions. Moreover, ANN can deal with the noisy data. 

ANN architecture is ordered in three consecutive layers: input, hidden/s, and output. Every layer possesses a number of neurons, a weight, a bias, and output [19]. Initially, one must figure out all the variables, with the effect on the main process being variable. The data collection, normally established before the ANN steps, becomes the mirror of the problem area. The best ANN architecture is subjected to learning quality and generalization ability, which relies on whether the collected data fall within the variation margin of the variables and are big enough in size [8]. 

The type of the task to be handled by the ANN is crucial in finding the best architecture. For better performance of ANNs, the parameters such as the number of neurons in the hidden layer(s), number of the hidden layers, the momentum, and the learning rates should be optimized.

The performance of an ANN model in portending the output can be checked and assessed by five statistical correlations [3,5,7,10,20,21]:(1)$$\mathrm{Average}\;\mathrm{correlation}\;\mathrm{factor}\;\left(R^{2}\right)=1-\frac{\sum {\left({\left(W\;\%\right)}_{est}-{\left(W\;\%\right)}_{\exp}\right)}^{2}}{\sum {\left({\left(W\;\%\right)}_{est}-\bar{{\left(W\;\%\right)}_{\exp}}\right)}^{2}}$$
(2)$$\mathrm{Root}\;\mathrm{mean}\;\mathrm{square}\;\mathrm{error}\;\left(\mathrm{RMSE}\right)=\sqrt{\frac{1}{N}\;\sum {\left({\left(W\;\%\right)}_{est}-{\left(W\;\%\right)}_{\exp}\right)}^{2}}$$
(3)$$\mathrm{Mean}\;\mathrm{absolute}\;\mathrm{error}\;\left(\mathrm{MAE}\right)=\frac{1}{N}\sum \left|{\left(W\;\%\right)}_{est}-{\left(W\;\%\right)}_{\exp}\right|$$
(4)$$\mathrm{Mean}\;\mathrm{bias}\;\mathrm{error}\;\left(\mathrm{MBE}\right)=\frac{1}{N}\sum ({\left(W\;\%\right)}_{est}-\left(W\;\%{)}_{\exp}\right)$$
(5)$$\mathrm{Correlation}\;\mathrm{coefficient}\;\left(R\right)=\frac{\sum_{m=1}^{n}\left({\left(W\;\%\right)}_{\exp,m}-\bar{{\left(W\;\%\right)}_{\exp,m}}\right)\left({\left(W\;\%\right)}_{est,m}-\bar{{\left(W\;\%\right)}_{est,m}}\right)}{\sqrt{\sum_{m=1}^{n}{\left({\left(W\;\%\right)}_{\exp,m}-\bar{{\left(W\;\%\right)}_{\exp,m}}\right)}^{2}\sum_{m=1}^{n}{\left({\left(W\;\%\right)}_{est,m}-\bar{{\left(W\;\%\right)}_{est,m}}\right)}^{2}}}$$
where

*(W %)_est_*: is the estimated value of the weight left % by ANN model;*(W %)_exp_*, is the experimental value of the weight left %; and$\bar{\left(W\;\%\right)}$: is the average values of weight left %.

In order to get the best ANN model, it should be targeted to get the lowest error with (RMSE, MAE, MBE), and the highest with (R^2^, R) correlations [10]. In this investigation, weight left % of mixed polymers has been predicted by an ANN model. There are some advantages and some disadvantages for using ANN. Some of these advantages can be summarized as being easy to work with linear and non-linear relationships and learning these relationships directly from the data used, while a disadvantages is that doing the fitting needs big memory and computational efforts [22].

## 3. Results and Discussion 

### 3.1. TGA of Mixed Polymers 

TGA provides us with the thermogravimetric (TG), and the derivative thermogravimetric (DTG) at different heating rates of the pyrolysis of two sets at different polymers compositions, which are shown in Figure 1 and Figure 2, respectively [16]. 

### 3.2. Pyrolysis Prediction by ANN Model

Neural Network with “Feed-Forward, Back-Propagation” (FFBPNN) was established in “nntool” function in MATLAB^®^ R2020a based on 358, 752 data for the two sets. This type of ANN model is widely used because it is very efficient and simple [3]. Usually, in the thermal analysis instrument TGA, the raw signal (weight left %) will be the output of the ANN model and the independent variables (temperature and heating rate in the non-isothermal TGA data) could be the inputs of the ANN model.

The collected data will be divided by three subsets: training set will be used to establish the network learning and correct the weights by minimizing the error function; the validation set checks the performance of the network; and finally, the test set will test the generalization of the network [23]. 

The whole data comprising 358, 752 sets are shown in Table 3, and randomly divided into three sets as follows: 70% for training, 15% for validation and testing. Osman and Aggour [24] mentioned that collecting large sets of data could help the model with high accuracy. 

Table 4 listed the parameters of the ANN “nntool” model and Table 5 shows a comparison of different ANN structure performance with different numbers of hidden layers and different numbers of neuron and transfer functions in each hidden layer. Usually, the best architecture is found by a trial and error process [8]. The value of R is examined as the criteria in judging the most efficient network architecture for finding the percentage weight loss %. Values of four statistical correlations will be tabulated only for the last best-selected architecture. 

The final and best ANN architecture is AN7-A and AN7-B, as shown in Figure 3 for both sets. This network is utilized for the next simulation step. This architecture constitutes 10 neurons with logsig-tansig functions in the two hidden layers with linear transfer function for the output layer. Hidden layers with non-linear functions were used to deal with complex functions [2]. Usually, linear function is not recommended in the hidden layers in order to avoid a linearly separable prediction, while tansig is more preferable since it has larger range of output [11]. Most of the researchers mentioned in Table 1 implied more than one hidden layer [11]. The number of neurons in the hidden layer is a crucial parameter in the efficiency and the accuracy of the ANN output. To avoid the underfitting and the overfitting (too many neurons), one should select the number of neurons in such a way that the performance function will get eventually the optimum value [6,23,25]. There are different supervised learning algorithms such as Levenberg–Marquardt (LM), Bayesian Regularization, and Scaled Conjugate Gradient, but LM is used due its best performance and relevance for this data number [8,10,26]. This optimization LM algorithm technique will update the values of the weighted and biases factors in order to get the calculated output close to the target [5,10]. 

Figure 4 shows all the results fall close to the diagonal, which confirms a strong agreement and good correlation for ANN prediction with experimental values at minimum mean square error (MSE) values of 2.1275 × 10^−7^ and 4.58 × 10^−8^ of the two sets, respectively (Figure 5). This MSE’s values are too small (<2.1275 × 10^−7^), which shows that the prediction of the system is very reliable [8]. Naqvi et al. [8] also pointed out that for a good prediction ANN, output values should be close to the target values, and ANN model is a good fit for TGA data. 

The performance of the current AN7-A and AN7-B model in predicting the weight left % was measured by calculating these four statistical correlations. Table 6 shows all these four statistical correlations. Notice that values of RMSE, MAE, and MBE are significantly low. Consequently, this model can powerfully predict the output within an acceptable limit of error. 

Once checking the ANN for the two sets, the final architecture will be simulated by new input data. Table 7 presented the simulation stage with nine datasets for each AN7-A and AN7-B for only new input data, and the final network will produce the simulated output according to the final architecture AN7-A and AN7-B. Figure 6 shows the comparison between the simulated network with the actual output and indicates very high performance of the selected network. In addition, Table 8 lists all statistical parameters for each set: AN7-A and AN7-B. As presented, the value of R is slightly high (>0.9900) and RMSE, MAE, and MBE have reasonably low values. Finally, Figure 7 shows the error histogram for the two sets, which is distributed across the zero error normally [11]. The error lies in a very small value range (−0.00085 to 0.002678) for the first set and (−0.00123 to 0.000489) for the second set, which indicates very good performance of the proposed ANN model. 

## 4. Conclusions

Thermal cracking of polymers, consisting of PS, LDPE, and PP, was implemented using TGA at heating rate range (5–40 K/min), with two groups of sets: (ratio 1:1) a mixture of PS and PP, and (ratio 1:1:1) a mixture of PS, LDPE, and PP. TGA data are used in modeling ANN for two sets of PS, LDPE, and PP polymers in order to predict the weight left %. 

However, an efficient ANN model has been created to predict the thermal decomposition of these two sets separately. The best architecture of 2-10-10-1 (*logsig-tansig-purelin*) transfer functions has been adopted as the highest efficient model. This could foresee the output very precisely with high regression coefficient value. After that, the best model has been simulated with untrained input data, and its behavior (calculated output) shows a close agreement with the actual values (high R > 0.9999).

## Figures and Tables

**Figure 1 polymers-14-02638-f001:**
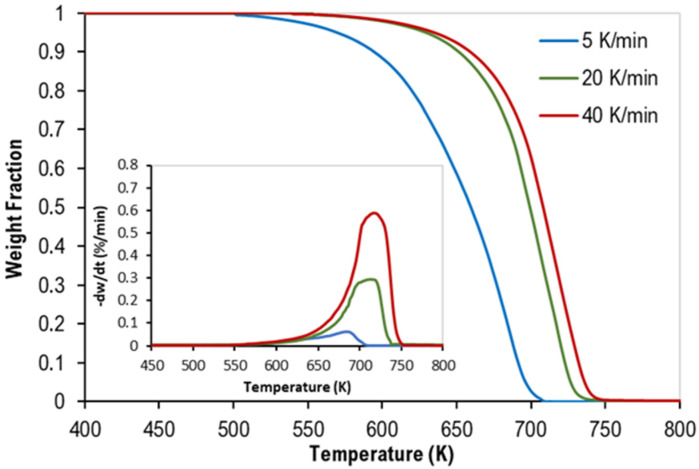
TG curves of binary mixtures of PP and PS with DTG curves inside.

**Figure 2 polymers-14-02638-f002:**
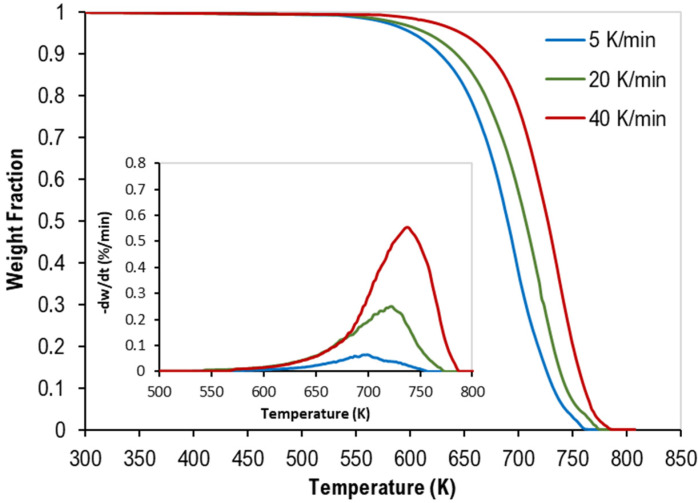
TG curves of ternary mixtures of PP, PS, and LDPE with DTG curves inside.

**Figure 3 polymers-14-02638-f003:**
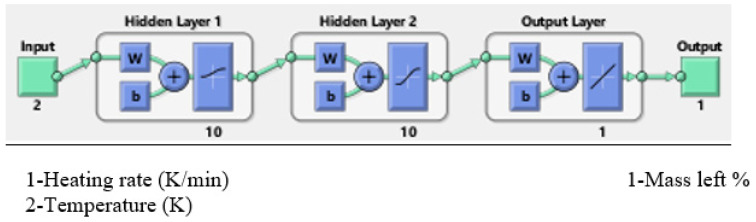
Topology of the selected AN7-A and AN7-B network.

**Figure 4 polymers-14-02638-f004:**
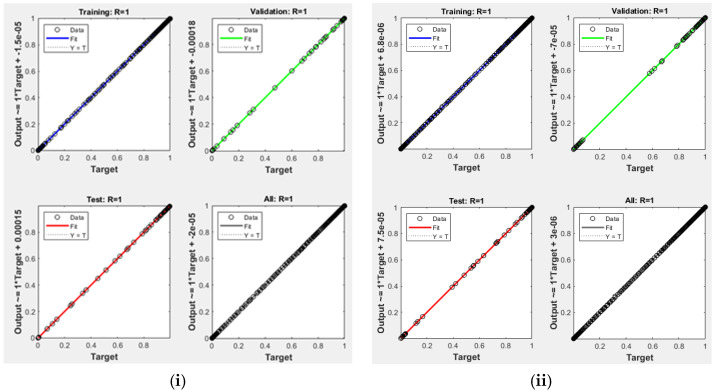
Regression of training, validation, and test plots for the selected (**i**) AN7-A, (**ii**) AN7-B.

**Figure 5 polymers-14-02638-f005:**
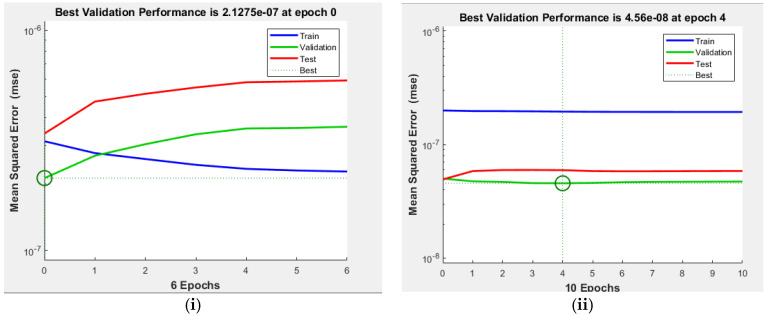
Mean square error for training, validation, and test plots for the selected (**i**) AN7-A, (**ii**) AN7-B.

**Figure 6 polymers-14-02638-f006:**
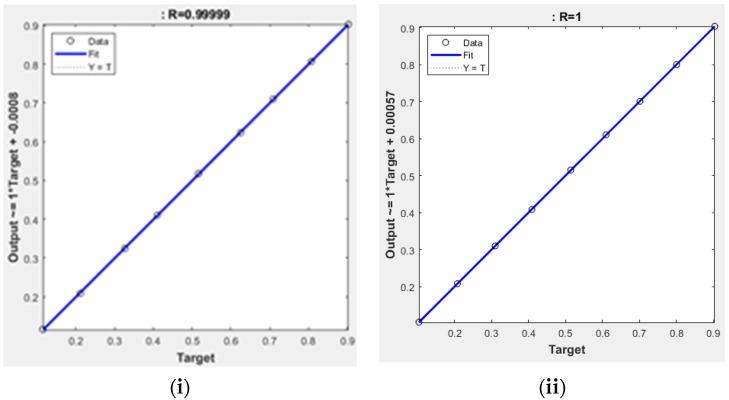
Regression of simulated data for (**i**) AN7-A, (**ii**) AN7-B.

**Figure 7 polymers-14-02638-f007:**
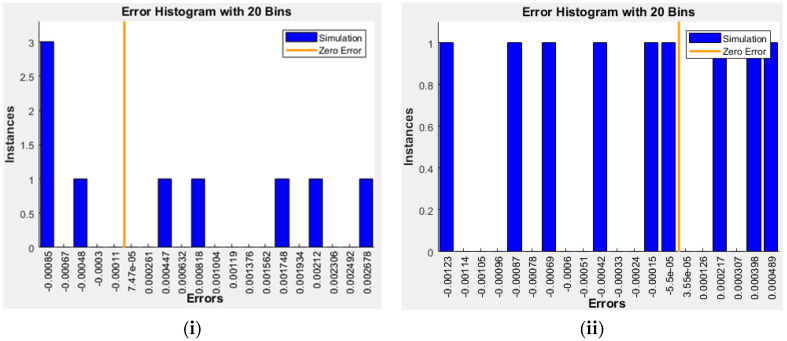
Error histogram of simulated data for (**i**) AN7-A, (**ii**) AN7-B.

**Table 1 polymers-14-02638-t001:** Literature summary of ANN applications for non-isothermal TGA data.

Author	Input Variables			Output Variables	Architecture Model	No. of Hidden Layers	Transfer Function for Hidden Layers	Data Points
Bezerra et al. [2]	temperature	heating rate	-	mass retained	2-21-21-1	2		1941
Yıldız et al. [3]	temperature	heating rate	blend ratio	Mass loss %	3-5-15-1	2	tangsig-tansig	
Ahmad et al. [5]	temperature	Heating rate	-	weight loss		2		1021
Çepelioĝullar et al. [6] Individual	temperature	heating rate	-	weight loss	2-20–20-1 (LFR)2-19–16-1 (OOR)	2	tangsig-logsig	4000
Çepelioĝullar et al. [6] Combined					2-7–6-1	2		8000
Chen et al. [7]	temperature	heating rate	mixing ratio	mass loss %	3-3-19-1	2	tansig-tansig	
Naqvi et al. [8]	temperature	heating rate	-	weight loss	2-5-1	1	tansig	1400
Ahmad et al. [9]	temperature	Heating rate	-	weight loss	2-10-1	1		1155
Bi et al. [10] (combustion), (pyrolysis)	temperature	mixing ratio	-	residual mass	2-3-18-1 2-3-15-1	2	tangsig-tangsig	
Bong et al. [11]	temperature	heating rate	-	weight loss %	2-(9-12)-(9-12)-1	2	tansig-tansig and logsig-tansig	
Bi et al. [12]	temperature	heating rate	mixing ratio	remaining mass %	3-5-10-1	2	tangsig-tangsig	5000
Zaker et al. [14]	temperature	heating rate	-	weight loss (%)	2-7-1	1	tansig	
Al-Yaari and Dubdub [17]	temperature	heating rate	mass ratio	mass left %	3-10-10-1	2	tansig-logsig	900

**Table 2 polymers-14-02638-t002:** List of six runs of different PS, LDPE, and PP polymers compositions.

Set No.	Test No.	Heating Rate (K/min)	Weight %			Comment
			PP	PS	LDPE	
1	1	5	50	50	0	mixture of PS, and PP
	2	20	50	50	0	
	3	40	50	50	0	
2	4	5	33.3	33.3	33.3	mixture of PS, LDPE, and PP
	5	20	33.3	33.3	33.3	
	6	40	33.3	33.3	33.3	

**Table 3 polymers-14-02638-t003:** Data set number of six tests.

Set No.	Test No.	Heating Rate (K/min)	Data Set Number	Total
1	1	5	126	358
	2	20	101	
	3	40	131	
2	4	5	251	752
	5	20	251	
	6	40	250	

**Table 4 polymers-14-02638-t004:** Main parameters of the ANN “nntool” model.

Number of inputs	2 (Temperature (K), Heating rate (K/min)
Number of output	1 (Mass left %)
Number of hidden layers	1-2
Transfer function of hidden layers	logsig-tansig
Number of neurons of hidden layers Transfer function of out layer	10-10 purelin
Data division function	Dividerand
Learning algorithm	Levenberg-Marquardt (TRAINLM)
Data division (Training-Validation-Testing)	70%-15%-15%
Data number (Training-Validation-Testing)	250-54-54 = 358 526-113-113 = 752
Data number (Simulation)	9-9
Performance function	MSE
Validation checks	6

**Table 5 polymers-14-02638-t005:** Comparison between different ANN structures for the two sets: (i) mixtures of PS and PP, (ii) mixtures of PS, LDPE, and PP.

Model	Network Topology (no. of Neurons) 2 Input-Hidden Layers (1 or 2 Layers)-1 Output	Hidden Layers		R
		1st Transfer Function	2nd Transfer Function	
i				
AN1-A	2-5-1	*tansig*	-	0.99881
AN2-A	2-5-1	*logsig*	-	0.99972
AN3-A	2-10-1	*tansig*	-	0.99995
AN4-A	2-10-1	*logsig*	-	0.99997
AN5-A	2-15-1	*tansig*	-	0.99997
AN6-A	2-15-1	*logsig*	-	0.99999
**AN7-A**	**2-10-10-1**	** *logsig* **	** *tansig* **	**1.00000**
ii				
AN1-B	2-5-1	*tansig*	-	0.99976
AN2-B	2-5-1	*logsig*	-	0.99997
AN3-B	2-10-1	*tansig*	-	0.99999
AN4-B	2-10-1	*logsig*	-	0.99999
AN5-B	2-15-1	*tansig*	-	0.99999
AN6-B	2-15-1	*logsig*	-	0.99999
**AN7-B**	**2-10-10-1**	** *logsig* **	** *tansig* **	**1.00000**

**Table 6 polymers-14-02638-t006:** Statistical parameters of the (A) AN7-A, (B) AN7-B model.

Set	AN7-A				AN7-B			
	Statistical Parameters				Statistical Parameters			
	R^2^	RMSE	MAE	MBE	R^2^	RMSE	MAE	MBE
Training	1.0	0.00055	0.00030	−0.00001	1.0	0.00044	0.00016	1.49 × 10^−6^
Validation	1.0	0.00046	0.00029	−0.00001	1.0	0.00021	0.00012	−1.74 × 10^−6^
Test	1.0	0.00058	0.00032	0.000018	1.0	0.00024	0.00014	0.000034
**All**	**1.0**	**0.00054**	**0.00030**	**−0.000012**	**1.0**	**0.000389**	**0.000154**	**6.018 × 10^−6^**

**Table 7 polymers-14-02638-t007:** Simulation input data and output data: mixtures of PS and PP mixtures of PS, LDPE, and PP.

No.	Mixture of PS and PP for AN7-A			Mixture of PS, LDPE, and PP for AN7-B		
	Input Data		Output Data	Input Data		Output Data
	Heating Rate (K/min)	Temperature (K)	Weight Fraction	Heating Rate (K/min)	Temperature (K)	Weight Fraction
1	5	690	0.11471	5	731	0.10335
2	5	668	0.41012	5	697	0.40892
3	5	634	0.70892	5	669	0.70090
4	20	716	0.21154	20	731	0.20736
5	20	698	0.51639	20	705	0.51387
6	20	672	0.80757	20	669	0.80014
7	40	718	0.32648	40	741	0.30962
8	40	700	0.62535	40	717	0.60931
9	40	658	0.90289	40	671	0.90323

**Table 8 polymers-14-02638-t008:** Statistical parameters for the simulated data of AN7-A and AN7-B.

AN7-A				AN7-B			
Statistical Parameters				Statistical Parameters			
R^2^	RMSE	MAE	MBE	R^2^	RMSE	MAE	MBE
0.99999	0.00144	0.00123	−0.00052	0.99999	0.00062	0.00049	0.00026

## Data Availability

Not applicable.

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
