# Peer review of "Pyrolysis Study of Mixed Polymers for Non-Isothermal TGA: Artificial Neural Networks Application"

_polymers, 2022, doi:10.3390/polym14132638_

Round 1

Reviewer 1 Report

In this contribution, the author utilized the artificial neuron network (ANN) to simulate the pyrolysis of polymer mixtures. This highlight ANN employs a 2-10-10-1 architecture, comprising temperature and heating rates as two input variables, 10 neurons in logsig and tansig transfer functions for two hidden layers, and left mass as the output. This ANN predicts values showing good coincidences with experimental results. This work is for the readership of Polymers, and the results consolidate the author’s proposal. Thus, I would recommend its publication after a revision addressing the following comments.

1.    Although the author mentioned selecting an appropriate neutron number avoids underfitting or overfitting (Line 157), the advantages of 10 neutrons, as well as 2 hidden layers, need more elaboration. 

2.    What is tested in Table 8 needs more details, though “all statistical parameters for each set” is mentioned in Line 193. Does Table 8 present predicted values of a new TGA dataset?

3.    In the insets of Figure 1 and 2, the values and units on vertical axes seem incorrect. For example, in Figure 2, the 40 K/min weight fraction curve shows a -dw/dt of above 50 (%/min) near 738 K, instead of ~0.56 (%/min) near the maximum in the inset. 

4.    In Line 96, the author introduces that the performance of an ANN model is assessed by four statistical correlations, but five correlations are listed in Line 97-102. Later, Table 6 and 8 present four correlations except for the average correlation factor, R^2. Please check if R^2 in Line 97 is necessary.

5.    Figure 1, 2 and 3 need appropriate citations for reuse. Besides, Figure 3 needs a higher resolution for better legibility.

6.    In Line 155, the author introduces “Most of researchers mentioned in Table 2 implied more than one hidden layer”. This statement refers to Table 1 instead of Table 2.

Reviewer 2 Report

The paper is interesting and the results of the described research may be useful in the design of plastic pyrolysis technology.  

However, I consider it necessery to complete part 2.1.  of the description of device in which thermal decomposition process was carried out.
